# Screening for *α*-Glucosidase-Inhibiting Saponins from Pressurized Hot Water Extracts of Quinoa Husks

**DOI:** 10.3390/foods11193026

**Published:** 2022-09-29

**Authors:** Rong Su, Jing Li, Na Hu, Honglun Wang, Jingya Cao, Xiaofeng Chi, Qi Dong

**Affiliations:** 1Medical College of Qinghai University, Xining 810016, China; 2CAS Key Laboratory of Tibetan Medicine Research, Northwest Institute of Plateau Biology, Xining 810008, China

**Keywords:** *Chenopodium quinoa* Willd, pressurized hot water extraction, response surface methodology, saponins, *α*-glucosidase, molecular docking

## Abstract

The present study extracted total saponins from quinoa husks with pressurized hot water extraction and optimized the extraction conditions. The response surface methodology (RSM) with a Box–Behnken design (BBD) was employed to investigate the effects of extraction flow rate, extraction temperature and extraction time on the extraction yield of total saponins. A maximal yield of 23.06 mg/g was obtained at conditions of 2 mL/min, 210 °C and 50 min. The constituents of the extracts were analyzed by liquid chromatography–mass spectrometry (LC-MS). A total of twenty-three compounds were identified, including five flavonoids, seventeen triterpenoid saponins and a phenolic acid. Moreover, we performed an in vitro assay for the *α*-glucosidase activity and found a stronger inhibitory effect of the quinoa husk extracts than acarbose, suggesting its potential to be developed into functional products with hypoglycemic effect. Finally, our molecular docking analyses indicated triterpenoid saponins as the main bioactive components.

## 1. Introduction

Quinoa (*Chenopodium quinoa* Willd.) is an annual crop of the genus Chenopodiaceae [1], originating from the Andes of South America. It may help solve food shortage problems due to its strong tolerance and growth adaptability to cold, drought, saline–alkali, barren and other severe soil and climate conditions [2]. As one of the top healthy foods globally, quinoa has high nutritional and biological values and is known as the “golden grain” [3]. In addition to the large number of nutrients it contains [4], quinoa is also rich in saponins, phenols, polysaccharides and other phytochemicals [5,6,7] that exert excellent health-protecting effects, including anti-inflammatory, antibacterial, antioxidant and antidiabetic properties [8].

As food, quinoa grains are usually dehulled. Quinoa husks are mostly discarded [9], which causes a waste of resources and pollution. Interestingly, studies have shown that quinoa husks are a good source of saponins [10]. In fact, the majority of saponins in quinoa husks are triterpenoid glycosides, which are derived from β-vanillin and comprise sugar chains and sapogenins [11]. Quinoa saponins possess a variety of biological and physiological properties, such as hypoglycemia, antioxidation, anti-cancer and anti-inflammation [12,13,14,15]. Moreover, studies also suggest that the hypoglycemic effect of triterpenoid saponins is mainly due to their inhibition of *α*-glucosidase and improved *β*-cell function and insulin resistance [16,17]. Therefore, it is significant to explore a safe and efficient extraction method for quinoa saponins and develop the related health-benefiting functional products to promote the maximal utilization of quinoa husks.

Pressurized hot water extraction (PHWE) is an environmentally friendly and efficient extraction technology using water as the extractant. The water remains in a liquid state when the temperature rises to 100–374 °C under the critical pressure (1–22.1 MPa) [18,19]. Under such state, the dielectric constant (*ε*) of water significantly decreases with increased temperature, which makes the properties of the water similar to those of organic solvents and thus capable of extracting compounds of low polarity [20]. Compared with traditional extraction methods, pressurized hot water extraction has the advantages of short extraction time, high extraction efficiency, safety and environmental friendliness [21], and it has been used in the extraction of polyphenols [22,23,24], polysaccharides [25,26] and other natural products. However, few studies have been conducted on the effects of pressurized hot water extraction on the yield of total saponins from quinoa husks.

The present study investigated the effects of PHWE extraction parameters (extraction flow rate, extraction temperature and extraction time) on the yield of total saponins using the response surface methodology (RSM) with a central composite design. Meanwhile, we also analyzed the chemical constituents of the extracts (by LC-MS), their potential inhibition on *α*-glucosidase (using an in vitro glycosidase activity assay) and the molecular interactions (by molecular docking).

## 2. Materials and Methods

### 2.1. Materials

The powder of the husks of *Chenopodium quinoa* was provided by Qinghai Boji Biotechnology Co., Ltd. (Xining, China). *Saccharomyces cerevisiae*-derived *α*-glucosidase (G5003) was purchased from Sigma-Aldrich (Shanghai, China). p-Nitrophenyl-*α*-d-glucopyranoside (pNPG) was purchased from Aladdin Reagents (Shanghai, China). Acarbose was purchased from Shanghai yuanye Bio-Technology Co., Ltd. (Shanghai, China). The other reagents and solvents used were of analytical grade.

### 2.2. Optimization of the Extraction of Total Saponins

#### 2.2.1. Experimental Design

A three-level–three-factor Box–Behnken design response surface methodology (BBD-RSM) [27] was utilized to optimize the conditions to maximize the yield of total saponins from quinoa husks. The extraction flow rate (A, mL/min), extraction temperature (B, °C) and extraction time (C, min) were set as factors. Three levels were considered for each factor (Table 1).

#### 2.2.2. Pressurized Hot Water Extraction of Total Saponins from Quinoa Husks

The pressurized hot water extraction was performed with a laboratory-built system (constructed by the Northwest Institute of Plateau Biology, Chinese Academy of Sciences, China). The extraction system consisted of a liquid chromatography infusion pump (LC-20AT, SHIMADZU, Kyoto, Japan) delivering ultra-pure water, a gas chromatography column oven (GC-14B, Shanghai Kechuang Chromatography Instruments, Shanghai, China) controlling the extraction temperature, a back pressure regulator (TESCOM, Elk River, MN, USA), SS-0006WT-B-P micro-channel heat exchanger (Hangzhou Shenshi Energy Conservation Technology Co., Ltd., Hangzhou, China), stainless coil preheater (0.3 mm in inner diameter and 2.0 m in length) and a 100 cm^3^ stainless steel reaction vessel (2.0 cm in diameter and 8.0 cm in height, internal dimensions). The powder of quinoa husks (1.000 g) mixed with 8–16 mesh quartz sand was placed into the reaction vessel. The ultra-pure water was delivered into the stainless coil preheater in the column oven by the pump and previously heated, and when the temperature reached the set point, the ultra-pure water was pumped into the extraction vessel at a constant flow rate. The extraction procedure was carried out according to the conditions in Table 2. The extraction liquid was filtered, concentrated and dried.

#### 2.2.3. Determination of the Content of Total Saponins

The extraction yield of total saponins was measured following a previously described method, with modifications [28]. Firstly, 0.2 mL of quinoa husk extracts was evaporated to dryness in a 70 °C water bath and cooled down. Amounts of 0.4 mL of 5% vanillin-acetic acid solution and 1.6 mL of 72% perchloric acid were added in sequence and incubated in a 70 °C water bath for 10 min, followed by cooling with ice-cold water for another 10 min. Then, the mixture was mixed with 10 mL of 17 M acetic acid and incubated at 25 °C for 15 min. The absorbance was measured at 560 nm using a Cary 300 Bio spectrometer (Agilent Corporation, Santa Clara, CA, USA), with the oleanolic acid solution as standard and the sample solution as a blank control.

#### 2.2.4. Optimization of RSM

RSM was used to determine the optimal extraction conditions for total saponins from quinoa husks. The extraction flow rate, extraction temperature and extraction time were set as the investigation factors, and the yield of total saponins was set as the response value. The Box–Behnken center combination design was carried out by RSM, with the data being analyzed with Design Expert 8.0.

### 2.3. LC-MS Analysis of Quinoa Husk Extracts at Optimal Extraction Conditions

The chemical constituents of the extracts at the optimal extraction conditions were analyzed by LC-MS. The extracts (1 mg) added to 1 mL of 80% methanol were sonicated and centrifuged to obtain the supernatant. The UPLC-Triple-TOF-MS system (Waters Corp., Milford, USA) was employed for the analysis of components. The ultra HPLC system was equipped with a HSS T3 column (1.7 μm, 2.1 mm × 150 mm), which was used at the column temperature of 50 °C. The mobile phase was composed of solvent A (0.1% formic acid) and solvent B (0.1% formic acid acetonitrile), and gradient elution (0–10 min, 95–75% A; 10–25 min, 75–40% A; 25–32 min, 40–5% A) was performed. The flow rate was 0.3 mL/min, the volume loaded was 2 μL, and the detection wavelength was 230 nm. The mass spectrometry analysis was performed with a scan range of 100–1500 *m/z* in the negative ion scan mode. The mass spectrometry data were acquired under the following conditions: nebulizer gas (GS1): 55 psi; nebulizer gas (GS2): 55 psi; curtain gas (CUR): 35 psi; ion source temperature (TEM): 550 °C (negative), 600 °C (positive); ion source voltage (IS): −4500 V (negative), 5500 V (positive); primary scan: declustering voltage (DP): 100 V; focusing voltage (CE): 10V; secondary scan: TOF MS~Product Ion~IDA mode, the CID energy was 20, 40 and 60 V. The mass axis was calibrated with a CDS pump with an error <2 ppm.

### 2.4. Measurement of α-Glucosidase Activity of Quinoa Husk Extracts at Optimal Extraction Conditions

In vitro measurement of the *α*-glucosidase activity was performed as previously described, with modifications [29]. The test included a blank group, a sample group and a sample control group. Amounts of 20 μL of the sample (quinoa husk extracts solution) of different concentrations and 80 μL of 0.1 mM phosphate-buffered solution (PBS, pH 6.8) were added to a 96-well plate. The sample control group was added with 50 μL of 0.1 mM PBS (pH 6.8), and the other groups were added with 50 μL of *α*-glucosidase solution (1 U/mL). After incubating for 30 min at 37 °C, 50 μL of 0.5 mM pNPG solution was added and mixed well and incubated at 37 °C for 30 min. Finally, the reaction was stopped by adding 50 μL of 0.1 mol/L Na_2_CO_3_ solution. The absorbance value was measured at 405 nm (EPOCH2 microplate reader, BioTek, Inc., Vermont, USA). Acarbose was used as the positive control. The inhibition rate of each sample was calculated according to the following formula.
$$\mathrm{Inhibition}\;\left(\%\right)=\frac{A-(A_{1}-A_{2})}{A}\times 100$$
A: Absorbance value of blank group; A_1_: Absorbance value of sample group; A_2_: Absorbance value of sample control group.

### 2.5. Molecular Docking

Molecular docking was performed to determine the interaction between the *α*-glucosidase and the constituents in the quinoa husk extracts. As the 3D structure of *α*-glucosidase was unavailable, the isomaltose from *Saccharomyces cerevisiae* with high sequence homology to α-glucosidase was used for the docking analysis. The crystal structure of isomaltose was downloaded from the Protein databank (PDB ID: 3A4A). A stable receptor was created with AutoDock by adding hydrogen atoms and removing water and ligands to modify the protein [30]. ChemBio 3D Ultra 19.0 was used to create the 3D structures of the compounds and standard acarbose, and the energy minimization was performed to obtain a reasonable conformation. Then, the AutoDock Vina [31] software was used to dock the modified molecular structures and the protein crystal structure. Compounds with lower binding energy were considered to have stronger *α*-glucosidase inhibitory activity, and some compounds with better docking scores were selected to analyze the docking interactions.

### 2.6. Statistical Analysis

Experiments were performed in triplicate. Data were expressed as mean ± standard deviation. The results of response surface methodology were analyzed and graphed using Design Expert 8.0.

## 3. Results and Discussion

### 3.1. Optimization of the Extraction Conditions for Total Saponins with RSM

According to the BBD-RSM with three levels and three factors, the factors and levels of the response surface are shown in Table 1. Taking the extraction yield of total saponins (Y) as the response value, the Box–Behnken experimental design and results are shown in Table 2. Regression analysis was performed on the data in Table 2 using Design Expert 8.0, and the response surface variance analysis and results obtained are shown in Appendix A.

Based on statistical analysis of the experimental data, the relationships between the yield of total saponins and extraction parameters were described by the following equation:

Y = 19.10 + 0.49 × A + 1.16 × B + 0.063 × C − 1.01 × AB + 0.73 AC +0.82 × BC + 0.29 × A^2^ + 2.66 × B^2^ − 1.76 × C^2^


The response surface analysis of variance (ANOVA) showed a high significance of the regression (*p* < 0.01), whereas the lack of fit was not significant (*p* > 0.05), implying a high accuracy and reliability of our model (Appendix A). The F value represented the effects of extraction flow rate, extraction temperature and extraction time on the yield of total saponins, with a higher F value representing a greater effect [32]. Within the range of the selected factors, their F-value-based ranking is listed in Appendix A, showing an order of extraction temperature > extraction flow rate > extraction time. Therefore, the effect of extraction temperature on the yield of total saponins from quinoa husks is most significant.

The interaction diagram of the extraction flow rate, extraction temperature and extraction time was plotted according to the regression equation (Figure 1). A steep interaction curve meant a very significant effect of the interaction [33]. Combined with the results of the ANOVA, the interactions of two independent variables had significant effects on the yield of total saponins from quinoa husks, while extraction temperature was the most influential variable. The equation was solved by the regression model, and the optimal extraction conditions were as follows: extraction flow rate of 2.03 mL/min, extraction temperature of 209.94 °C and extraction time of 48.40 min. For convenience in practical operation, the conditions were modified to be 2.00 mL/min, 210 °C and 50 min. With this optimal condition, the yield of total saponins averaged from three independent experiments was 23.06 mg/g. The relative error for the predicted value (23.64 mg/g) was 2.5%, suggesting a high reliability of the regression model obtained by RSM.

It should be noted that the extraction temperature was a key parameter for the extraction yield of total saponins in this study. The effect might be attributed to the dielectric constant and polarity of the solvent, which could decrease along with the rising temperature and promote the dissolution of the saponins component, improving the yield. Additionally, with increased temperature, the movement and material exchange between saponin molecules accelerate; the viscosity and surface tension of the solvent reduce and thereby improve the wetting, penetrating and dissolving ability of the solvent [34]. However, more studies are required to determine whether more mechanisms are involved in the effect of temperature on extraction yield.

Previous studies with ultrasonic-assisted extraction, microwave-assisted extraction, supercritical CO_2_ extraction and solvent reflux extraction reported a yield of total saponins as 23.7 mg/g [35], 26.32 mg/g [36], 9.6 mg/g [37] and 16.85 mg/g [38], respectively. The values were comparable to those obtained by pressurized hot water extraction in this study (Table 3). However, compared with previous extraction methods, the pressurized hot water extraction method was more environmentally friendly and safer. For instance, the successful extraction of hydrophobic saponins with water reduces the usage of organic solvents.

### 3.2. Identification of Chemical Constituents in Quinoa Husk Extracts

The total ion flow diagram of chemical constituents in quinoa husk extracts by LC-MS is shown in Figure 2. Based on the molecular ion, main fragments and retention time in the MS spectrum, a total of twenty-three compounds were identified (Figure 3 and Figure 4), including five flavonoids, seventeen triterpenoid saponins and a phenolic acid, the main information on which is listed in Table 4.

Compounds **1**–**5** and **8** are phenols, mainly flavonoids. In positive and negative ion modes, flavonoid glycosides are prone to cleavage, resulting in loss of the neutral fragments and cracking of the glycosidic bonds, generating the corresponding aglycone [45]. The sugar groups that make up flavonoid glycosides are mainly glucose, rhamnose, rutinose and galactose, with the corresponding main fragment ions of [M−H−162]^−^, [M−H−146]^−^, [M−H−308]^−^ and [M−H−162]^−^. Compounds **1**, **3** and **8** should contain quercetin nucleus because of the existence of fragment ion at *m/z* 301 [46]. Compound **1**, with ion [M−H]^−^ *m/z* 755.2144 and the secondary fragment ion *m/z* 300.0274, was speculated to be a quercetin derivative. Compared with the aglycone ion [Y_0_]^−^ *m/z* 301.0353, the relative abundance of the radical aglycone ion [Y_0_−H]^−^ *m/z* 300.0274 was higher, indicating that it may be quercetin 3-*O*-glycoside compound aglycone [40]. Overall, it was very likely that the molecular ion *m/z* 755.2144 lost two pentose molecules and a hexose molecule to produce the aglycone ion *m/z* 300.0274. In sum, Compound **1** should be quercetin 3-*O*-(2,6-di-*α*-l-rhamnopyranosyl)-*β*-d-galactopyranoside. Compound **3** should be quercetin 3-*O*-*β*-d-glucuronopyranoside due to the existence of fragment [M−H−179]^−^ generated by missing one molecule of glucuronic acid. Compound **8** was identified as quercetin because of the [M−H]^−^ ion at *m/z* 301.0366 and the fragment ion *m/z* 151.0033 generated by retro-Diels–Alder cleavage after losing the neutral fragment CO [47]. The presence of the kaempferol parent nucleus fragments ion at *m/z* 285 suggests Compound **2** is a kaempferol derivative. Its ion [M−H]^−^ *m/z* 739.2213, by the loss of two rhamnose molecules and one glucose molecule to obtain aglycone, further suggests it is kaempferol 3-*O*-(2,6-di-*α*-l-rhamnopyranosyl)-*β*-d-galactopyranoside. Compound **4** was preliminarily identified as myricetin-3-*o*-*β*-d-galactopyranoside [40], with the ion [M−H]^−^ at *m/z* 479.3041; it could lose a hexoside to produce the secondary fragment ion at *m/z* 319.1914, which corresponded to myricetin aglycones. The deduced structures of Compounds **1**–**5** and **8** are shown in Figure 3. Compound **4** was identified for the first time from quinoa, while the other compounds were identified from *Chenopodium quinoa* [39,41].

Compounds **6**–**7** and **9**–**23** are triterpenoid saponins, in which the aglycones (oleanolic acid, hederagenin, phytolaccagenic acid, serjanic acid) are linked with glycosides (glucose, galactose, arabinose, glucuronic acid, xylose) at C-3 or C-28 [10]. The triterpenoid saponins in quinoa mainly exist in the form of oxyglycosides, which undergo glycosidic bond cleavage in the positive or negative ion modes and generate glycosyl fragment ions ([M−H−162]^−^, [M−H−132]^−^, [M−H−179]^−^) and aglycone ions [48]. The existence of ions [M−H]^−^ at *m/z* 455 suggests Compounds **14**, **16**, **19**, **21**, **22** and **23** contain oleanolic aglycone and were characterized as oleanolic acid saponins. Compounds **14** and **19** were assigned as 3-*O*-*β*-d-glucuronopyranosyl oleanolic acid 28-*O*-*β*-glucopyranosyl ester ([M−H]^−^ ion at *m/z* 793.4496) and 3-*O*-*β*-d-glucuronopyranosyl oleanolic acid ([M−H]^−^ ion at *m/z* 631.3916), respectively. Both compounds had the glucuronic acid representative fragment [M−H−179]^−^, with the difference in Compound **14** showing fragment ion at *m/z* 631.3915 caused by the loss of a hexoside from *m/z* 793.4496. It was worth noting that Compound **21** depicted similar fragment ions to Compound **14**; they were likely to be isomers. Compound **22**, with deprotonated molecule ion [M−H]^−^ at *m/z* 763.4042 and aglycone ion *m/z* 455.3562 generated by losing xylose and glucuronic acid [44], was preliminarily speculated to be 3-*O*-*β*-d-xylopyranosyl-(1-3)-*β*-d glucuronopyranosyl oleanolic acid. Compound **23** had similar fragment ions to Compound **22**, likely to be isomers with each other. Compounds **11**, **12**, **13**, **18** and **20** all had fragment ions at *m/z* 471, indicating the presence of hederagenin aglycone. Compound **11** exhibited the deprotonated molecule ion [M−H]^−^ at *m/z* 809.4465, the ion *m/z* 647.3887 generated by its loss of a hexoside and the ion *m/z* 471.3520 by a further loss of a glucuronic acid, whose fragmentation pattern corresponded to the previously reported 3-O-*β*-d-glucuronopyranosyl hederagenin 28-O-*β*-d-glucopyranosyl ester saponin [42]. It is reported that a compound of large molecular size could lose its acidic group to generate the fragment ion [M+HCOOH−H]^−^ [39], which is the case for the carboxyl-groups-containing Compounds **12** and **13**. Compound **12**, with ion *m/z* 973.5057, lost fragment [M−H−44−162]^−^ and generated fragment ion *m/z* 765.4548, which further generated hederagenin aglycone ion *m/z* 471.3520 by losing a glucose molecule and a xylose molecule; it could be identified as hederagenin 3-*O*-[*β*-d-glucopyranosyl-(1,3)-*α*-l-arabinopyranosyl]-28-*O*-*β*-d- glucopyranoside by comparison with previous studies [39]. Compound **13** was characterized as basellasaponin A ([M−H]^−^ ion at *m/z* 969.4519), and the fragment ion at *m/z* 809.4471 represented the [M−H−44−116]^−^; the aglycone ion was generated by the loss of a glucose molecule and a glucuronic acid molecule. Compounds **9**, **10**, **15** and **17** were assigned as phytolaccagenic acid saponins because of the aglycone fragment ions at *m/z* 515; they all contained carboxyl groups, with the difference in positions where glycosides were attached. Compound **10** was speculated as 3-*O*-*β*-d-glucopyranosyl-(1-3)-*O*-*α*-l-arabinopyranosyl phytolaccagenic acid. It showed an ion at *m/z* 855.4429 and a fragment ion *m/z* 809.4522 due to losing a carboxyl group, with the latter further generating the aglycone fragment ion at 515.3412 by the loss of a glucose molecule and an arabinose molecule. Compound **15** was identified as 3-*O*-*α*-l-arabinopyranosyl phytolaccagenic acid ([M−H]^−^ ion at *m/z* 693.3514), which also produced a phytolaccagenic acid aglycone ion by decarboxylation and the removal of a pentose molecule. Compound **6** exhibited deprotonated molecule ion [M−H]^–^ at *m/z* 957.4882; the fragment ion at *m/z* 633.3719 suggested the loss of multiple glucose molecules, whereas the *m/z* 501.3251 indicated the missing of an arabinose molecule. Therefore, Compound **6** was deduced to be serjanic acid 3-*O*-[*β*-d-glucopyranosyl-(1-3)-*α*-l-arabinopyranosyl]-28-*O*-*β*-d-glucopyranoside, as previously reported in the analysis of seed hulls of quinoa [42]. The fragment ion at *m/z* 487 suggests Compound **7** was a pentacyclic triterpenoid saponin-containing entagenic acid. The ion *m/z* 781.4515 suggested the missing of the carboxyl fragment [M−H−44]^−^ from ion [M−H]^–^ *m/z* 827.4482, while the fragment ion at *m/z* 619.3924 and 487.3436 corresponded to the losses of a glucose molecule and an arabinose molecule, respectively. By referring to the previously reported structure [43], Compound **7** was deduced to be 3*β*,15*α*,16*α*-trihydroxy-18*β*-olean-12-en-28-oic acid 28-*O*-*α*-l-arabinopyanosyl-(1-3)-*β*-d-glucopyranosyl ester. The structures of Compounds **7**, **9**, **16**, **17**, **18** and **20** were firstly identified from *Chenopodium quinoa* and were not found in other natural products. The speculated structures of Compounds **6**–**7** and **9**–**23** are presented in Figure 4.

The present study identified a variety of flavonoids and triterpenoid saponins with the UPLC-Triple-TOF-MS system. The determination of the structures of these compounds with mass spectrometry provided a reference for a systematic and comprehensive understanding of the chemical components in quinoa husks. It could also be used to direct the separation and preparation of active ingredients from quinoa husks in the following studies. However, we realize the existence of limitations for this analytical method in determining the linkage positions of sugar moieties, and more methods will be employed to confirm the structures in our future studies.

### 3.3. Inhibitory Effect of Quinoa Husk Extracts on α-Glucosidase

#### 3.3.1. In Vitro Inhibition of α-Glucosidase Activity by Quinoa Husk Extracts

The potential inhibition of *α*-glucosidase by quinoa husk extracts at the optimal extraction conditions was measured using pNPG as substrate. The commercially available *α*-glucosidase inhibitor acarbose served as a positive control and was frequently used as a reference standard for evaluating the effect of candidate *α*-glucosidase inhibitors [49].

Figure 5 shows the inhibition rate of different concentrations of quinoa husk extracts and acarbose on *α*-glucosidase. The results illustrated that along with the increase in their concentrations, the inhibition rate of *α*-glucosidase increased in a concentration-dependent manner. The quinoa husk extracts showed a lower IC_50_ value of 32.62 mg/mL compared to that of acarbose (64.71 mg/mL), suggesting a stronger inhibition on *α*-glucosidase (*p* < 0.05). Our finding is consistent with the previous reports of the strong effect of triterpenoid saponins and flavonoids in inhibiting the activity of *α*-glucosidase [50,51]. Moreover, studies also suggest the functions of key groups in the structure of flavonoids and triterpenoid saponins in their *α*-glucosidase inhibiting activity, such as the number and position of hydroxyl groups and the carbonyl group in flavonoids [52] and the position of sugar moieties in the aglycon in triterpenoid saponins [53]. Therefore, quinoa husk extracts possess potential inhibitory effects on *α*-glucosidase, the main active ingredients of which are probably flavonoids and triterpenoid saponins. Nevertheless, most *α*-glucosidase inhibitor screening studies select enzyme preparations from nonmammalian sources (usually *Saccharomyces cerevisiae*), which could present some limitations due to the difficulty in assessing the actual in vivo hypoglycemic potential [54]. Accordingly, in future research work, we will further verify the potential hypoglycemic activity of the extracts through in vivo assays.

#### 3.3.2. Interaction between α-Glucosidase and Components in Quinoa Husk Extracts

Molecular docking was performed on the 23 compounds identified to explore the mechanism of the *α*-glucosidase inhibiting effects by the quinoa husk extracts. The docking affinity values between the compounds and *α*-glucosidase are shown in Table 5 and Appendix A. Remarkably, most compounds showed lower affinity values (within the range of −12.7 to −7.7 kcal/mol) than acarbose, suggesting greater binding with *α*-glucosidase. The strong binding found here is in agreement with previous reports on the analysis of flavonoids and triterpenoid saponins interacting with *α*-glucosidase [55,56].

The interactions of some flavonoids and triterpenoid saponins with better affinity values are shown in Figure 6 and Figure 7. The binding site on *α*-glucosidase by flavonoids **1**, **2**, **3**, **8** was similar to that of acarbose [57]. They were stabilized at the active site of *α*-glucosidase and interacted with amino acid residues by hydrogen bonds, salt bridge, hydrophobic interaction and other types of forces, via residues, such as His280, Asp325, Asp307, Arg315, Leu313, Pro312, Thr310, Ser311, Asp242, Glu332 and Ala281. It was reported that a flavonoid with two catechol groups in rings A and B and a 3-OH group in the C ring was very likely to have stronger *α*-glucosidase inhibitory activity than acarbose [58], a notion, which is supported by our findings. For instance, the 5-OH, 7-OH and 5′-OH of Compound **3** could interact with amino acid residues Asp352, Gln353 and Asp242 through hydrogen bonds, which showed a better affinity value (−9.7 kcal/mol) compared to acarbose (−8.5 kcal/mol).

Triterpenoid saponins **9**, **14**, **16**, **19** exerted excellent binding affinity toward *α*-glucosidase. They could bind with amino acid residues in the catalytic active site of *α*-glucosidase through hydrogen bonds and hydrophobic interactions via amino acid residues, such as His280, Gly309, Asn 247, Ser240, Asp307, Pro312, Leu246, Val319, Val308 and Ala329. Both Compounds **14** and **19** contained glucuronic acid groups at the C-3 position, which formed hydrogen bonds with the amino acid residue Gly309 and Asn 247, respectively. Considering the higher affinity value of Compounds **14** and **19**, we speculate that hydrogen bonds serve as the main force to inhibit the catalytic activity. Moreover, the carboxyl group at the C-28 position may establish hydrogen bond to induce conformational change of *α*-glucosidase and thereby could reduce the catalytic activity [59]. Furthermore, the aglycones at C-28 connected to the glucose moiety could also effectively inhibit the *α*-glucosidase activity [60]. Taken together, the inhibitory effect on *α*-glucosidase is largely determined by the hydrogen bonds and hydrophobic interactions, which mainly depends on the nature of the interaction-related residues and that of the compounds themselves [61]. These results could expand our knowledge on the interaction mechanisms between *α*-glucosidase and triterpenoid saponins and facilitate the development of potential *α*-glucosidase inhibitors.

## 4. Conclusions

The present study utilized pressurized hot water extraction to extract total saponins from quinoa husks and optimized the extraction conditions to obtain a maximal yield of 23.06 mg/g. Meanwhile, twenty-three compounds, mainly triterpenoid saponins and flavonoids, were identified from the extracts by LC-MS. The in vitro glycosidase activity assay showed that the quinoa husk extracts possessed stronger inhibiting activity on *α*-glucosidase than acarbose, while molecular-docking-based interaction analyses further suggested the main bioactive components to be triterpenoid saponins. In conclusion, pressurized hot water extraction is an environmentally friendly, safe and efficient extraction method for quinoa husk saponins. Quinoa husk saponins are promising food supplements to control postprandial hyperglycemia and candidates to be developed into *α*-glucosidase inhibitors. In addition, the mechanism of the effects of quinoa husk saponins on hyperglycemia requires more studies to be elucidated.

## Figures and Tables

**Figure 1 foods-11-03026-f001:**
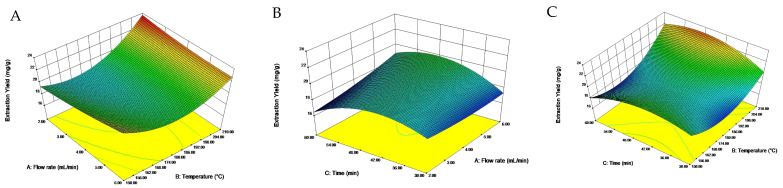
Interaction effects of factors on total saponins extraction yield in quinoa husks. (**A**) the interaction of flow rate and temperature; (**B**) the interaction of time and flow rate; (**C**) the interaction of time and temperature.

**Figure 2 foods-11-03026-f002:**
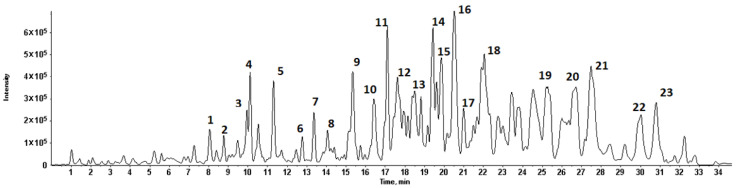
Total ion flow diagram of chemical constituents in quinoa husk extracts in negative ion mode by LC-MS.

**Figure 3 foods-11-03026-f003:**
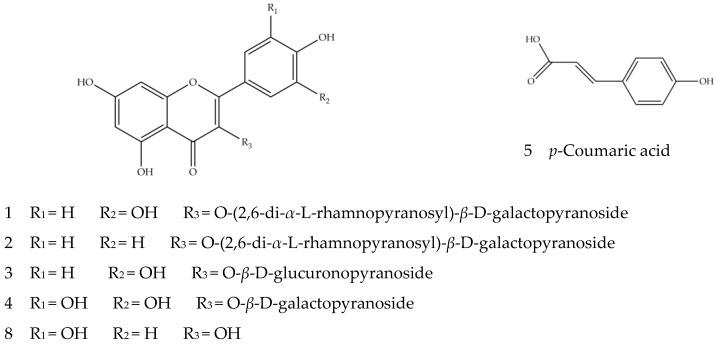
The structures of Compounds **1**–**5** and **8**.

**Figure 4 foods-11-03026-f004:**
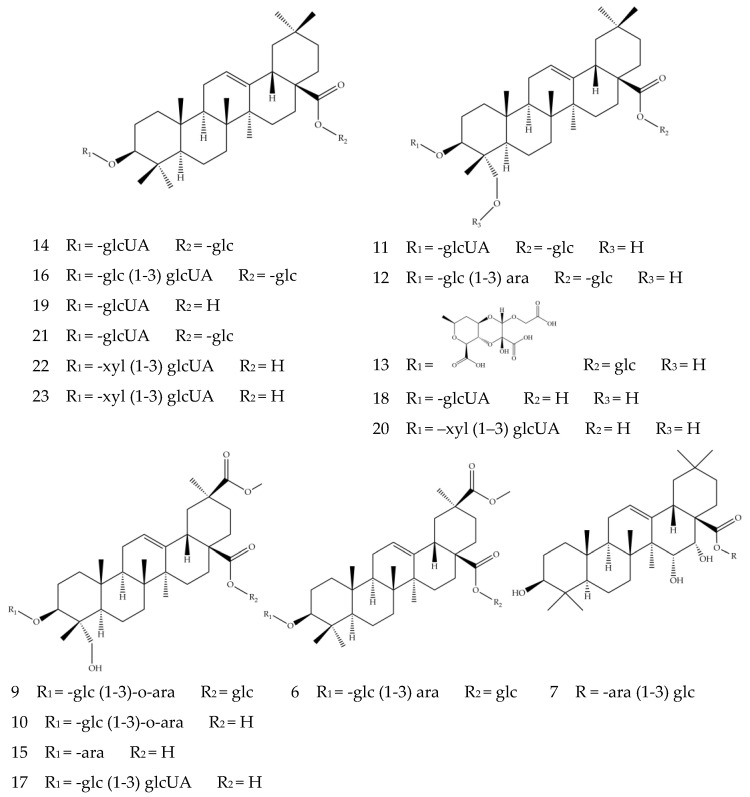
The structures of Compounds **6**–**7** and **9**–**23**.

**Figure 5 foods-11-03026-f005:**
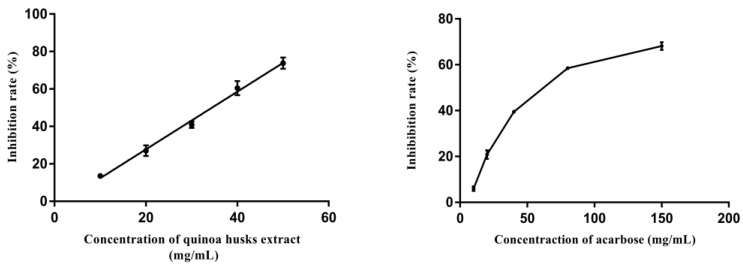
Inhibitory effect of the quinoa husk extracts and acarbose on *α*-glucosidase at different concentrations.

**Figure 6 foods-11-03026-f006:**
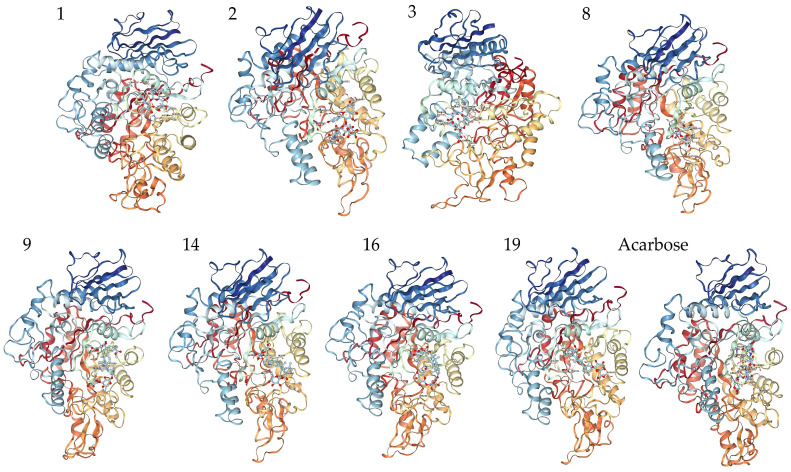
Molecular docking analysis and 3D modeling for the interaction between the compounds (Table 5) and *α*-glucosidase. Compounds **1**, **2**, **3**, **8** are flavonoids; Compounds **9**, **14**, **16**, **19** are triterpenoid saponins.

**Figure 7 foods-11-03026-f007:**
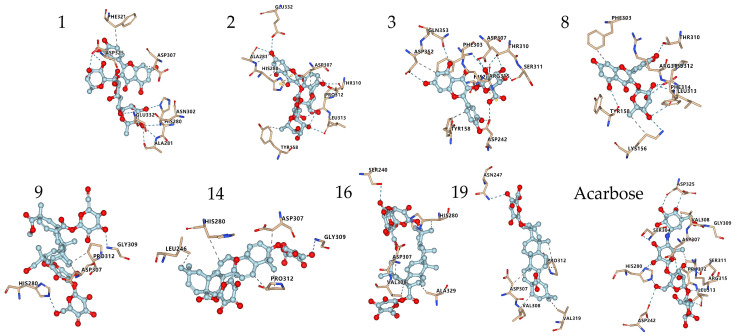
Conformations of the compounds (Table 5) interacting with amino acid residues at the active site of *α*-glucosidase. Compounds **1**, **2**, **3**, **8** are flavonoids; Compounds **9**, **14**, **16**, **19** are triterpenoid saponins.

**Table 1 foods-11-03026-t001:** Range and levels of experimental variables for the RSM.

Factors	Symbols	Levels		
		−1	0	1
Extraction Flow Rate (mL/min)	A	2	4	6
Extraction Temperature (°C)	B	150	180	210
Extraction Time (min)	C	30	45	60

**Table 2 foods-11-03026-t002:** Box–Behnken experimental design and results.

Run Order	Extraction Flow Rate (mL/min) A	Extraction Temperature (°C) B	Extraction Time (min) C	The Yield of Total Saponins (mg/g) Y
1	2	180	60	16.77 ± 0.56
2	4	180	45	18.67 ± 1.47
3	2	150	45	19.16± 0.38
4	4	210	60	21.89 ± 1.81
5	4	180	45	19.61 ± 1.05
6	6	180	30	17.02 ± 0.30
7	6	180	60	18.81 ± 0.18
8	6	150	45	22.54 ± 0.67
9	6	210	45	22.90 ± 1.29
10	4	180	45	19.01 ± 0.75
11	2	210	45	23.57 ± 0.69
12	2	180	30	17.88 ± 0.83
13	4	180	45	19.31 ± 1.28
14	4	150	30	19.74 ± 0.64
15	4	150	60	18.01 ± 0.29
16	4	180	45	18.91 ± 1.53
17	4	210	30	20.34 ± 1.58

**Table 3 foods-11-03026-t003:** Comparison of total saponins extraction technology in quinoa husks.

Extraction Technique	Solvent Used	Extraction Temperature (°C)	Extraction Time (min)	Solid/Solvent	Other Parameters	The Yield of Total Saponins (mg/g)	Ref
Ultrasonic-assisted extraction	75% EtOH	45	90	1:15	-	23.7	[35]
Microwave-assisted extraction	68% EtOH	-	10	1:32	Power 455 W	26.32	[35]
Supercritical CO_2_ extraction	74% EtOH	60	96	-	Pressure 37 MPa	9.6	[37]
Solvent reflux extraction	72% EtOH	72	147	1:20.8	-	16.85	[38]
Pressurized hot water extraction	Water	210	50	-	Flow rate 2 mL/min	23.06	

**Table 4 foods-11-03026-t004:** Twenty-three compounds identified from the extracts of quinoa husks by LC-MS.

NO.	RT (min)	[M-H] (*m/z*)	MS/MS Fragments	Formula	Compound	Ref
1	8.04	755.2144	755.2144, 300.0274.	C_33_H_40_O_20_	Quercetin 3-*O*-(2,6-di-*α*-l-rhamnopyranosyl) -*β*-d-galactopyranoside	[39]
2	8.77	739.2213	739.2213, 285.0417.	C_33_H_40_O_19_	kaempferol 3-*O*-(2,6-di-*α*-l-rhamnopyranosyl) -*β*-d-galactopyranoside	[39]
3	9.95	477.0687	477.0687, 301.0370.	C_21_H_18_O_13_	quercetin 3-*O*-*β*-d-glucuronopyranoside	[39]
4	10.11	479.3041	479.3041, 319.1914, 159.1016.	C_21_H_20_O_13_	Myricetin-3-*O*-*β*-d-galactopyranoside	[40]
5	11.30	187.0096	187.0096, 123.0821, 97.0676	C_9_H_8_O_3_	*p*-Coumaric acid	[41]
6	12.76	957.4882	957.4882, 795.4309, 633.3719, 501.3251,	C_48_H_76_O_19_	Serjanic acid 3-*O*-[*β*-d-glucopyranosyl-(1-3)-*α*-l- arabinopyranosyl]-28-*O*-*β*-d-glucopyranoside	[42]
7	13.36	827.4482 [M+COOH]^−^	827.4482, 781.4515, 619.3924, 487.3436	C_41_H_66_O_14_	3*β*,15*α*,16*α*-trihydroxy-18*β*-olean-12-en-28-oic acid 28-*O*-*α*-l-arabinopyanosyl-(1-3)-*β*-d- glucopyranosyl ester	[43]
8	14.05	301.0366	301.0366, 151.0033	C_15_H_10_O_7_	Quercetin	[41]
9	15.34	1017.4968 [M+COOH]^−^	855.4428, 809.4467, 647.3884, 515.3427	C_48_H_76_O_20_	3-*O*-*β*-d-glucopyranosyl-(1-3)-*O*-*α*-l-arabinopyranosyl phytolaccagenic acid 28-*O*-*β*-d- glucopyranosyl ester	-
10	16.42	855.4429 [M+COOH]^−^	855.4429, 809.4522, 647.3887, 515.3412,	C_42_H_66_O_15_	*O*-*β*-d-glucopyranosyl-(1-3)-O-*α*-l- arabinopyranosyl phytolaccagenic acid	[39]
11	17.10	809.4465	809.4465, 647.3887, 471.3520	C_42_H_66_O_15_	3-*O*-*β*-d-glucuronopyranosyl hederagenin 28-*O*-*β*-d-glucopyranosyl ester	[42]
12	17.63	973.5057 [M+COOH]^−^	973.5057, 765.4548, 603.3971, 471.3520	C_47_H_76_O_18_	Hederagenin 3-*O*-[*β*-d-glucopyranosyl-(1,3)-*α*- l-arabinopyranosyl]-28-*O*-*β*-d-glucopyranoside	[39]
13	18.49	969.4519	969.4519, 925.4610, 809.4471, 471.3521	C_47_H_70_O_21_	basellasaponin A	[39]
14	19.41	793.4496	793.4496, 631.3915, 455.3551	C_42_H_66_O_14_	3-*O*-*β*-d-glucuronopyranosyl oleanolic acid 28-*O*-*β*-d-glucopyranosyl ester	[42]
15	19.85	693.3514	693.3514, 647.3458, 515.3437	C_36_H_56_O_10_	3-*O*-*α*-l-arabinopyranosyl phytolaccagenic acid	[39]
16	20.51	953.4559	953.4559, 909.4650, 793.450, 631.3938, 455.3565	C_48_H_76_O_19_	*O*-*β*-d-glucopyranosyl-(1-3)-*β*-d- glucuronopyranosyl oleanolic acid 28-*O*-*β*-d-glucopyranosyl ester	-
17	20.99	851.4580	851.4580, 807.3955, 691.3794, 515.3426	C_43_H_66_O_17_	3-*O*-*β*-d-glucopyranosyl-(1-3)-*β*-d glucuronopyranosyl phytolaccagenic acid	-
18	22.04	647.3820	647.3820, 471.3506	C_36_H_56_O_10_	Hederagenin 3-*O*-*β*-d-glucuronopyranoside	-
19	25.32	631.3916	631.3916, 455.3558	C_36_H_56_O_9_	3-*O*-*β*-d-glucuronopyranosyl oleanolic acid	[44]
20	26.65	779.3988	779.3988, 647.3878, 471.3512	C_41_H_64_O_14_	Hederagenin 3-*O*-*β*-d-xylopyranosyl-(1-3)-*β*-d glucuronopyranoside	-
21	27.51	791.3892	791.3892, 631.3940, 455.3574	C_42_H_66_O_14_	14 isomer	[42]
22	29.97	763.4042	763.4042, 631.3930, 455.3562	C_41_H_64_O_13_	oleanolic acid 3-*O*-*β*-d-xylopyranosyl-(1-3)-*β*-d glucuronopyranoside	[44]
23	30.79	763.4042	763.4042, 631.3930, 455.3562	C_41_H_64_O_13_	22 isomer	[44]

**Table 5 foods-11-03026-t005:** Affinity values of major compounds with *α*-glucosidase.

	Flavonoids				Triterpenoid Saponins				Acarbose
Compound Number	1	2	3	8	9	14	16	19	
Affinity (kcal/mol)	−8.6	−8.1	−9.7	−8.9	−11.6	−12.6	−12.2	−12.7	−8.5

## Data Availability

All data are reported in this manuscript.

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
