# Peer review of "Screening for α-Glucosidase-Inhibiting Saponins from Pressurized Hot Water Extracts of Quinoa Husks"

_foods, 2022, doi:10.3390/foods11193026_

Round 1

Author Response

Thank you for your letter and for the reviewer’s comments concerning our manuscript entitled “Screening for α-Glucosidase-inhibiting Saponins from Pressurized Hot Water Extracts of Quinoa Husks”. ( ID: foods-1904440 ). Our deepest gratitude goes to the anonymous reviewer for careful work and thoughtful suggestions that have helped improve this manuscript substantially. Meanwhile, thank you very much for giving us a chance to revise and resubmit our manuscript.

Based on these comments, we have made careful modifications again. All changes made to the manuscript are marked in red. These changes will not influence the content and framework of the manuscript. We hope that these revisions are satisfactory and the revised manuscript will be acceptable for publication.

Below you will find our point-by-point responses to your comments. Once again, thank you very much for your comments and suggestions.

Best regards.

Reviewer 2 Report

Dear authors,

There is no doubt about the work behind the manuscript. I believe it is an interesting research work that must be shown to the scientific community. However, I would like you consider the following suggestions:

Format:

-Please, check the manuscript correcting those expression that must be in italic format. (i.e. m/z , in vitro and some scientific names)

Introduction

-line 33: It would be helpful to include some examples of mentioned health protecting effects.

--line 35-36: It is not a conclusion from the work relative to reference 9. What authors wrote is an assumption from such results so I suggest either rewrite the sentence or to include a more accurate reference.

-line 42: what do the authors mean when saying “…safer and more efficient extraction….” ? compare to what?

-“subcritical water extraction (SWE)” it is a controversial concept. Any condition bellows the critical point could directly be named “subcritical”, since no area in the pressure-temperature phase diagrams is defined for such state. I suggest using “pressurized hot water (PHW)” instead or/and “high-pressurized hot water”.

-line 55-56: please, check the reference https://doi.org/10.1016/j.indcrop.2018.04.074

Methods:

-2.2.2. Some relevant data are missing such as how the vessel was filled, the vessel volume, the residence time of the solvent, the temperature of the water while pumping before entering the vessel, how the collapses were avoided since the raw material was powdered, the total volume pumped by extraction which can have influence on mass transfer, etc

-How the authors corrected the interferences in the spectrophotometric analysis generated by pigments present in the extracts? No clarification step is detailed int the text.

-Why authors didn´t consider alpha glucosidase inhibition as a dependent variable together with yield? Wouldn’t be more interesting to obtain the optimal condition with higher yield and inhibition activity? (nevertheless, see below my comments about alpha-glucosidase inhibition )

Results:

Some figure legends are incomplete. Please include the information needed to make self-explanatory figures.

3.1.  Is it reliable the optimization considering that no clarification step for saponins quantification was performed?

3.2. I suggest avoiding the term “quasi-molecular ion”

- there are several errors related to a not found reference in this section

3.3.  No specification about which sample/s are tested for inhibition determination. Figure 5 is not clear, ¿to which sample is referred?

-The alpha glucosidases are in the brush border of the enterocytes so, when any material reaches the enzymes, it had been submitted to a digestion process. Therefore, when no digestion processes are included in bioactivities assays we must referred to “potential inhibition” since the influence of digestion process on the bioactive compounds under interest are unknown.

- Moreover,  the main concern about in vitro  alpha glucosidase inhibition chemical experiments (no cell culturing) is based on the reliability of the results because as Miller and Joubert (2021) DOI:10.1055/a-1557-7379 pointed, a nonmammalian enzyme was used in this work and there is strong evidence that inhibition data obtained using nonmammalian α-glucosidase may hold limited value in terms of identifying α-glucosidase inhibitors with actual in vivo hypoglycaemic potential, despite the inconsistencies when acarbose is used as reference standard.

It seems that authors pointed the triterpenoid saponins and flavonoids as main responsible of such bioactivity. The contribution of such groups of compounds (i.e. % as ug triterpenoids saponins/ 100 mg of extract) was not calculated so it not fully correct to perform such assumptions.  In fact, such contribution must be calculated to correlate the inhibition to any of the mentioned group of compounds. Therefore, I suggest the authors to complete the experimental work.

Moreover, considering that no clarification step was performed and that other compounds are present in the extracts (as total yields of saponins denoted), the inhibition cannot be directly attributed to saponins since other compounds could be potentially responsible.

This section must be deeply improved, mostly because the titled claimed that saponins are responsible of such bioactivity and following present results, such conclusion cannot be done.

I encouraged the author to consider my suggestions for manuscript improvement.

Best regards

Author Response

(The authors gave the same response as above.)

Round 2

Reviewer 2 Report

Dear authors:

Thank you for considering my comments. As I can see, the manuscript has been improved however, I would like to emphasize the relevance of points 8, 16 and 17 of your response to reviewer comments letter.

Point 8. When using sample solution without the other components to start the reaction, it must be called sample blank which give us an idea about the interference provided by the sample components without reaction (which is needed). The concern comes once within the sample there are compounds under interest (saponins) and other compounds that contribute in a whole to the “color” development with and/or without reaction. For such unspecific spectrophotometry methods, how do we know that the signal at 560 nm exclusively comes from the saponins and not from a combination between saponins+other compounds (i.e. pigments)? That is the reason a clarification step using a chloroform-butanol-water system may result crucial to assign any bioactivity to saponins content. However, because of the extraction methodology, this step may not be mandatory for all plant materials, which can be your case. So, I would make sure by testing at least one sample that no interferences occur by the other compounds presence by comparing your data with a one experiment including clarification step. Nevertheless, it is the responsibility of the authors that the saponin quantification data are reliable so, once the explanation was made, if the authors properly justify that it is not necessary , it is fine for me.

Point 16. I completely agree the authors about how widely used is such methodology and the limitations to improve it. However, I would mention such limitations on the text noting that you are aware.

Point 17. The author´s answer is reasonable so, even I would include such extra data and some explanation considering what was discussed in point 8 softening the statements performed, I have nothing else to add.

Best regards

Author Response

Thanks for your letter and for reviewer’s comments concerning our manuscript entitled “Screening for α-Glucosidase-inhibiting Saponins from Pressurized Hot Water Extracts of Quinoa Husks”. (Manuscript ID: foods-1904440 ). Those comments are all valuable and helpful for revising and improving our paper. Meanwhile, thank you very much for giving us a chance to revise and resubmit our manuscript. We have studied all the comments carefully and have made conscientious correction. Revised portion are marked in red in the manuscript. These changes will not influence the content and framework of the manuscript. We appreciate for the reviewer’ warm work earnestly, and hope that these correction will meet with approval.

Below you will find our point-by-point responses to your comments. Once again, thank you very much for your comments and suggestions.

Thank you and best regards.
